# Universal models for binary spike patterns using centered Dirichlet processes

**Il Memming Park**[123], **Evan Archer**[24], **Kenneth Latimer**[12], **Jonathan W. Pillow**[1234]
1. Institue for Neuroscience, 2. Center for Perceptual Systems, 3. Department of Psychology
4. Division of Statistics & Scientific Computation
The University of Texas at Austin
{memming@austin., earcher@, latimerk@, pillow@mail.} utexas.edu

## Abstract

Probabilistic models for binary spike patterns provide a powerful tool for un-
derstanding the statistical dependencies in large-scale neural recordings. Maxi-
mum entropy (or "maxent") models, which seek to explain dependencies in terms
of low-order interactions between neurons, have enjoyed remarkable success in
modeling such patterns, particularly for small groups of neurons. However, these
models are computationally intractable for large populations, and low-order max-
ent models have been shown to be inadequate for some datasets. To overcome
these limitations, we propose a family of "universal" models for binary spike pat-
terns, where universality refers to the ability to model arbitrary distributions over
all $2^m$ binary patterns. We construct universal models using a Dirichlet process
centered on a well-behaved parametric base measure, which naturally combines
the flexibility of a histogram and the parsimony of a parametric model. We derive
computationally efficient inference methods using Bernoulli and cascaded logis-
tic base measures, which scale tractably to large populations. We also establish a
condition for equivalence between the cascaded logistic and the 2nd-order maxent
or "Ising" model, making cascaded logistic a reasonable choice for base measure
in a universal model. We illustrate the performance of these models using neural
data.

## 1 Introduction

Probability distributions over spike words form the fundamental building blocks of the neural code.
Accurate estimates of these distributions are difficult to obtain in the context of modern experimen-
tal techniques, which make it possible to record the simultaneous spiking activity of hundreds of
neurons. These difficulties, both computational and statistical, arise fundamentally from the expo-
nential scaling (in population size) of the number of possible words a given population is capable
of expressing. One strategy for combating this combinatorial explosion is to introduce a parametric
model which seeks to make trade-offs between flexibility, computational expense [1, 2], or math-
ematical completeness [3] in order to be applicable to large-scale neural recordings. A variety of
parametric models have been proposed in the literature, including the 2nd-order maxent or Ising
model [4, 5], the reliable interaction model [3], restricted Boltzmann machine [6], deep learning [7],
mixture of Bernoulli model [8], and the dichotomized Gaussian model [9]. However, while the num-
ber of parameters in a model chosen from a given parametric family may increase with the number
of neurons, it cannot increase exponentially with the number of words. Thus, as the size of a popula-
tion increases, a parametric model rapidly loses flexibility in describing the full spike distribution. In
contrast, nonparametric models allow flexibility to grow with the amount of data [10, 11, 12, 13, 14].
A naive nonparametric model, such as the histogram of spike words, theoretically preserves repre-
sentational power and computational simplicity. Yet in practice, the empirical histogram may be
extremely slow to converge, especially for the high dimensional data we are primarily interested

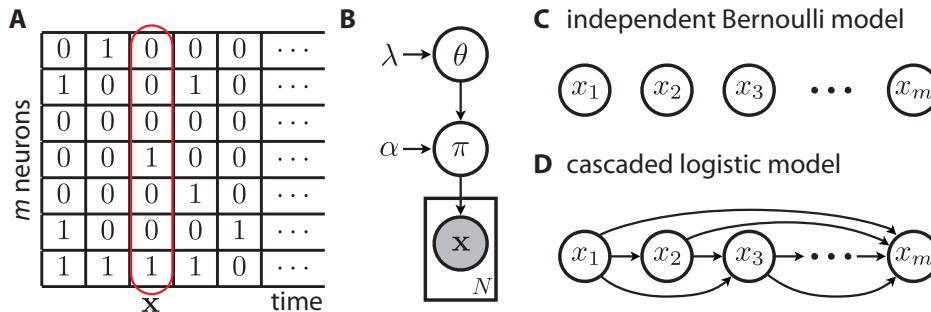

Figure 1: **(A)** Binary representation of neural population activity. A single spike word $\mathbf{x}$ is indicated in red. **(B)** Hierarchical Dirichlet process prior for the universal binary model (UBM) over spike words. Each word is drawn with probability $\pi_j$. The $\pi$'s are drawn from a Dirichlet with parameters given by $\alpha$ and a base distribution over spike words with parameter $\theta$. **(C, D)** Graphical models of two base measures over spike words: independent Bernoulli model and cascaded logistic model. The base measure is also a distribution over each spike word $\mathbf{x} = (x_1, \ldots, x_m)$.

in. In most cases, we expect never to have enough data for the empirical histogram to converge. Perhaps even more concerning is that a naive histogram model fails smooth over the space of words: unobserved words are not accounted for in the model.

We propose a framework which combines the parsimony of parametric models with the flexibility of nonparametric models. We model the spike word distribution as a Dirichlet process centered on a parametric base measure. An appropriately chosen base measure smooths the observations, while the Dirichlet process allows for data that depart systematically from the base measure. These models are *universal* in the sense that they can converge to any distribution supported on the $(2^m - 1)$-dimensional simplex. The influence of any base measure diminishes with increasing sample size, and the model ultimately converges to the empirical distribution function.

The choice of base measure influences the small-sample behavior and computational tractability of universal models, both of which are crucial for neural applications. We consider two base measures that exploit *a priori* knowledge about neural data while remaining computationally tractable for large populations: the independent Bernoulli spiking model, and the cascaded logistic model [15, 16]. Both the Bernoulli and cascaded logistic models show better performance when used as a base measure for a universal model than when used alone. We apply these models to several simulated and neural data examples.

## 2 Universal binary model

Consider a (random) binary spike word of length $m$, $\mathbf{x} \in \{0, 1\}^m$, where $m$ denotes the number of distinct neurons (and/or time bins; Fig. 1A). There are $K = 2^m$ possible words, which we index by $k \in \{1, \ldots, K\}$. The universal binary model is a hierarchical probabilistic model where on the bottom level (Fig. 1B), $\mathbf{x}$ is drawn from a multinomial (categorical) distribution with the probability of observing each word given by the vector $\boldsymbol{\pi}$ (spike word distribution). On the top level, we model $\boldsymbol{\pi}$ as a Dirichlet process [11] with a discrete base measure $G_\theta$, hence,

$$\mathbf{x} \sim \mathrm{Cat}(\boldsymbol{\pi}), \qquad \boldsymbol{\pi} \sim \mathrm{DP}(\alpha G_\theta), \qquad \theta \sim p(\theta|\lambda), \tag{1}$$

where $\alpha$ is the *concentration* parameter, $G_\theta$ is the *base measure*, a discrete probability distribution over spike words, parameterized by $\theta$, and $p(\theta|\lambda)$ is the hyper-prior. We choose a discrete probability measure for $G_\theta$ such that it has positive measure only over $\{1, \ldots, K\}$, and denote $g_k = G_\theta(k)$. Thus, the Dirichlet process has probability mass only on the $K$ spike words, and is described by a (finite dimensional) Dirichlet distribution,

$$\boldsymbol{\pi} \sim \mathrm{Dir}(\alpha g_1, \ldots, \alpha g_K). \tag{2}$$

In the absence of data, the parametric base measure controls the mean of this nonparametric model,

$$\mathrm{E}[\boldsymbol{\pi}|\alpha] = G_\theta, \tag{3}$$

regardless of $\alpha$. Therefore, we loosely say that $\boldsymbol{\pi}$ is "centered" around $G_\theta$.[1] We can start with good parametric models of neural populations, and extend them into a nonparametric model by using them as the base measure [17]. Under this scheme, the base measure quickly learns much of the basic structure of the data while the Dirichlet extension takes into account any deviations in the data which are not predicted by the parametric component. We call such an extension a *universal binary model* (UBM) with base measure $G_\theta$.

The marginal distribution of a collection of words $X = \{\mathbf{x}_i\}_{i=1}^N$ under UBM is obtained by integrating over $\boldsymbol{\pi}$, and has the form of a Polya (a.k.a. Dirichlet-Multinomial) distribution:

$$P(X|\alpha, G_\theta) = \frac{\Gamma(\alpha)}{\Gamma(N+\alpha)} \prod_{k=1}^K \frac{\Gamma(n_k + \alpha g_k)}{\Gamma(\alpha g_k)}, \qquad (4)$$

where $n_k$ is the number of observations of the word $k$. This leads to a simple formula for sampling from the predictive distribution over words:

$$\Pr(x_{N+1} = k | X_N, \alpha, G_\theta) = \frac{n_k + \alpha g_k}{N + \alpha}. \qquad (5)$$

Thus, sampling proceeds exactly as in the Chinese restaurant process (CRP): we set the $(N+1)$-th word to be $k$ with probability proportional to $n_k + \alpha g_k$, and with probability proportional to $\alpha$ we draw a new word from $G_\theta$ (which in turn increases the probability of getting word $k$ on the next draw). Note that as $\alpha \to 0$, the predictive distribution converges to the histogram estimate $\frac{n_k}{N}$, and as $\alpha \to \infty$, it converges to the base measure itself. We use the Jensen-Shannon divergence to the predictive distribution to quantify the performance in our experiments.

## 2.1 Model fitting

Given data, we fit the UBM via maximum a posteriori (MAP) inference for $\alpha$ and $\theta$, using coordinate ascent. The marginal log-likelihood from (4) is given by,

$$L = \log P(X_N|\alpha, \theta) = \sum_k \log \Gamma(n_k + \alpha g_k) - \sum_k \log \Gamma(\alpha g_k) + \log \Gamma(\alpha) - \log \Gamma(N+\alpha). \quad (6)$$

Derivatives with respect to $\alpha$ and $\theta$ are,

$$\frac{\partial L}{\partial \theta} = \alpha \sum_k \left( \psi(n_k + \alpha g_k) - \psi(\alpha g_k) \right) \frac{\partial}{\partial \theta} g_k, \qquad (7)$$

$$\frac{\partial L}{\partial \alpha} = \sum_k g_k \left( \psi(n_k + \alpha g_k) - \psi(\alpha g_k) \right) + \psi(\alpha) - \psi(N+\alpha), \qquad (8)$$

where $\psi$ denotes the digamma function. Note that the summation terms vanish when we have no observations ($n_k = 0$), so we only need to consider the words observed in the dataset.

Note also that in the limit $\alpha \to \infty$, $\frac{\mathrm{d}L}{\mathrm{d}\theta}$ converges to $\sum \frac{n_k}{g_k} \frac{\partial}{\partial \theta} g_k$, the derivative of the logarithm of the base measure with respect to $\theta$. On the other hand, in the limit $\alpha \to 0$, the derivative goes to $\sum \frac{1}{g_k} \frac{\partial}{\partial \theta} g_k$, reflecting the fact that the number of observations $n_k$ is ignored: the likelihood effectively reflects only a single draw from the base distribution with probability $g_k$.

Even when the likelihood defined by the base measure is a convex or log-convex in $\theta$, the UBM likelihood is not guaranteed to be convex. Hence, we optimize by a coordinate ascent procedure that alternates between optimizing $\alpha$ and $\theta$.

## 2.2 Hyper-prior

When modeling large populations of neurons, the number of parameters $\theta$ of the base measure grows and over-fitting becomes a concern. Since the UBM relies on the base measure to provide smoothing over words, it is critical to properly regularize our estimate of $\theta$.

We place a hyper-prior $p(\theta|\lambda)$ on $\theta$ for regularization. We consider both $l_2$ and $l_1$ regularization, which correspond to Gaussian and double exponential priors, respectively. With regularization, the loss function for optimization is $L - \lambda \|\theta\|_p^p$, where $p = 1, 2$. In a typical multi-neuron recording, the connectivity is known to be sparse and lower order [1, 3], and so we assume the connectivity is sparse. The $l_1$ prior in particular promotes sparsity.

## 3 Base measures

The scalability of UBM hinges on the scalability of its base measure. We describe two computationally efficient base measures.

### 3.1 Independent Bernoulli model

We consider the independent Bernoulli model which assumes (statistically) independent spiking neurons. It is often used as a baseline model for its simplicity [4, 3]. The Bernoulli base measure takes the form,

$$G_\theta(k) = p(x_1, \ldots, x_m|\theta) = \prod_i^m p_i^{x_i}(1 - p_i)^{1-x_i}, \tag{9}$$

where $p_i \geq 0$ and $\theta = (p_1, \ldots, p_m)$. The distribution has full support on $K$ spike words as long as all $p_i$'s are non-zero. Although the Bernoulli model cannot capture the higher-order correlation structure of the spike word distribution with only $m$ parameters, inference is fast and memory-efficient.

### 3.2 Cascaded logistic model

To introduce a rich dependence structure among the neurons, we assume the joint firing probability of each neuron factors with a cascaded structure (see Fig. 1D):

$$p(x_1, x_2, \ldots, x_m) = p(x_1)p(x_2|x_1)p(x_3|x_1, x_2) \cdots p(x_m|x_1, x_2, \ldots, x_{m-1}). \tag{10}$$

Along with a parametric form of conditional distribution $p(x_i|x_1, \ldots, p_{i-1})$, it provides a probabilistic model of spike words.

A natural choice of the conditional is the logistic-Bernoulli linear model—a widely used model for binary observations [2].

$$p(x_i = 1|x_{1:i-1}, \theta) = logistic(h_i + \sum_{j<i} w_{ij}x_j) \tag{11}$$

where $\theta = (h_i, w_{ij})_{i,j<i}$ are the parameters. The combination of the factorization and the likelihoods give rise to the cascaded logistic (Bernoulli) model[2], which can be written as,

$$G_\theta(k) = p(x_1, \ldots, x_m|\theta) = \prod_{i=1}^m p(x_i|x_{1:i-1}) \tag{12}$$

$$p(x_i|x_{1:i-1}, \theta) = \left[1 + \exp\left(-(2x_i - 1)\left(h_i + \sum_{j=1}^{i-1} w_{ij}x_j\right)\right)\right]^{-1} \tag{13}$$

The cascaded logistic model and the Ising model (second order maxent model) have the same number of parameters $\frac{m(m+1)}{2}$, but a different parametric form. The Ising model can be written as[3],

$$p(x_1, \ldots, x_m|\theta) = \frac{1}{Z(J)} \exp\left(\sum_{i,j \leq i} J_{ij}x_ix_j\right) \tag{14}$$

where $\theta = J$ is a upper triangular matrix of parameters, and $Z(J)$ is the normalizer. However, unlike the cascaded logistic model, it is difficult to evaluate the likelihood of the Ising model, since it does not have a computationally tractable normalizer (partition function). Hence, fitting an Ising model is typically challenging. Since each conditional can be independently fit with a logistic regression (a

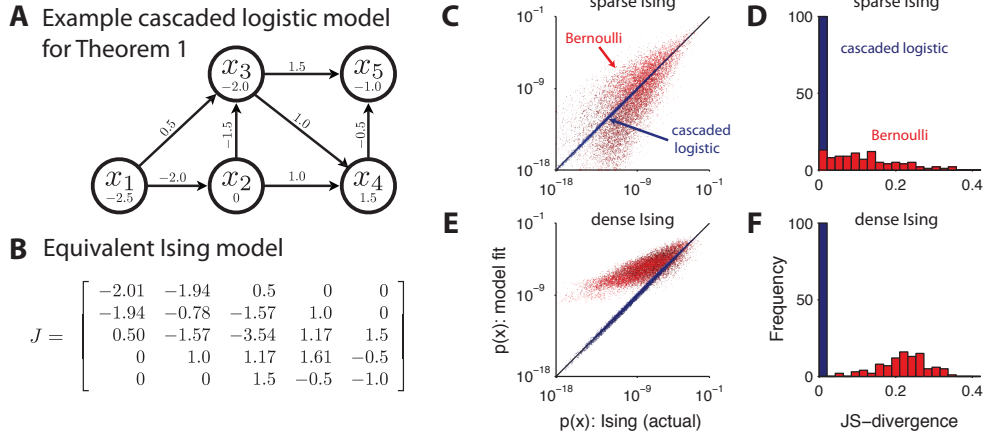

Figure 2: Tight relation between cascaded logistic model and the Ising model. **(A)** A cascaded logistic model depicted as a graphical model with at most two conditioning (incoming arrow) per node (see Theorem 2). The $h_i$ parameters are given in the nodes and the interaction terms, $w_{ij}$ are shown on the arrows between nodes. **(B)** Parameter matrix $J$ of an Ising model equivalent to (A). **(C)** A scatter plot of three simulated Ising models fit with cascaded logistic (blue tone) and independent Bernoulli (red tone) models. Each point is a word in the $m = 15$ spike word space. The x-axis gives probability of the word under the actual Ising model and the y-axis shows the estimated probability from the fit model. The Ising model parameters were sparsely connect and generated randomly. The diagonal terms ($J_{ii}$) were drawn from a standard normal. 80% of the off-diagonal ($J_{ij}, i \neq j$) terms were set to 0 and the rest drawn from a normal with mean 0 and standard deviation 3. Both models were fit by maximum likelihood using $10^7$ samples. **(D)** A histogram of the Jensen-Shannon (JS) divergence between 100 random pairs of sparse Ising model and the fit models. **(E,F)** Same as (C,D) for Ising models generated with dense connectivity. The diagonal terms in the Ising model parameters were constant -2. The off-diagonal terms were drawn from a standard normal distribution.

convex optimization), cascaded logistic model's estimation is computationally tractable for a large number of neurons [2].

Despite these differences, remarkably, the Ising model and the cascaded logistic models overlap substantially. Up to $m = 3$ neurons, Ising model and cascaded logistic model are equivalent. For larger populations, the following theorem describes the intersection of the two models.

**Theorem 1** (Pentadiagonal Ising model is a cascaded logistic model). *An Ising model with $J_{ij} = 0$ for $j < i-2$ or $j > i+2$, is also a cascaded logistic model. Moreover, the parameter transformation is bijective.*

The mapping between models parameters is given by

$$J_{m,m} = h_m \tag{15}$$

$$J_{m-1,m} = w_{m,m-1} \tag{16}$$

$$J_{m-1,m-1} = h_{m-1} + \log\left(\frac{1 + \exp(h_m)}{1 + \exp(h_m + w_{m,m-1})}\right) \tag{17}$$

$$J_{i,i} = h_i + \log\left(\frac{1 + \exp(h_{i+1})}{1 + \exp(h_{i+1} + w_{i+1,i})}\right) + \log\left(\frac{1 + \exp(h_{i+2})}{1 + \exp(h_{i+2} + w_{i+2,i})}\right) \tag{18}$$

$$J_{i,i+1} = w_{i+1,i} + \log\left(\frac{(1 + \exp(h_{i+2} + w_{i+2,i}))(1 + \exp(h_{i+2} + w_{i+2,i+1}))}{(1 + \exp(h_{i+2}))(1 + \exp(h_{i+2} + w_{i+2,i+1} + w_{i+2,i}))}\right) \tag{19}$$

$$J_{i,i+2} = w_{i+2,i} \tag{20}$$

for $1 \leq i \leq n - 2$, for a symmetric $J$. Proof can be found in the supplemental material.

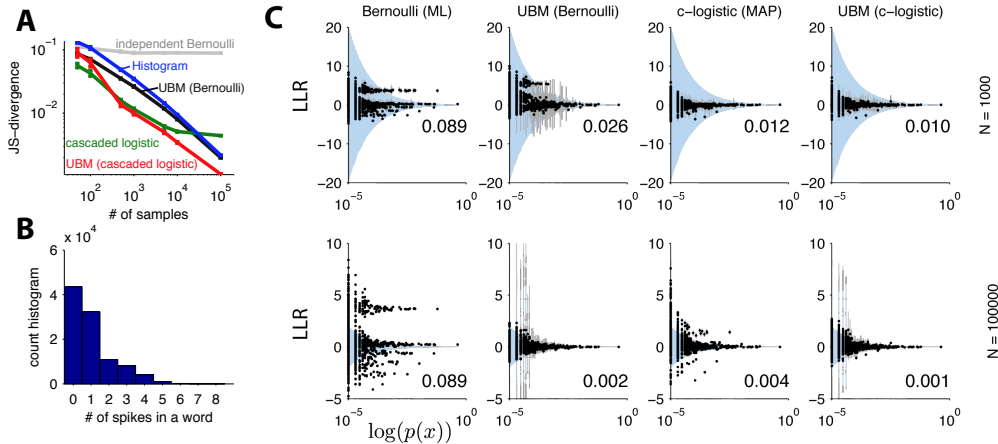

Figure 3:  3rd order maxent distribution experiment. **(A)** Convergence in Jensen-Shannon (JS) divergence between the fit model and the true model. Error bar represents SEM over 10 repeats. **(B)** Histogram of the number of spikes per word. **(C)** Scatter plots of the log-likelihood ratio $\log(P_{\mathrm{emp}}(k)) - \log(P_{\mathrm{model}}(k))$ for each model (column), and two sample sizes of $N = 1000$ and $N = 100000$ (rows). Note the scale difference on the y-axes. Error line represents twice the standard deviation over 10 repeats. Shaded area represents frequentist 95% confidence interval for histogram estimator assuming the same amount of data. The number on the bottom right is the JS divergence.

Unlike the Ising model, the order of the neurons plays a role in the formulation of the cascaded logistic model. Since a permutation of a pentadiagonal matrix is not necessarily pentadiagonal, this poses a potential challenge to the application of this equivalency. However, the Cuthill-McKee algorithm can be used as a heuristic to find a permutation of $J$ with the lowest bandwidth (i.e., closest to pentadiagonal) [18].

This theorem can be generalized to sparse, structured cascaded logistic models.

**Theorem 2** (Intersection between cascaded logistic model and Ising model). *A cascaded logistic model with at most two interactions with other neurons is also an Ising model.*

For example, cascaded logistic with a sparse cascade $p(x_1)p(x_2|x_1)p(x_3|x_1)p(x_4|x_1,x_3)p(x_5|x_2,x_4)$ is an Ising model (Fig. 2A)[4]. We remark that although the cascaded logistic model can be written as an exponential family form, the cascaded logistic does not correspond to a simple family of maximum entropy models in general.

The theorems show that only a subset of Ising models are equivalent to cascaded logistic models. However, cascaded logistic models generally provide good approximations to the Ising model. We demonstrate this by drawing random Ising models (both with sparse and dense pairwise coupling $J$), and then fitting with a cascaded logistic model (Fig. 2C-F). Since Ising models are widely accepted as effective models of neural populations, the cascaded logistic model presents a computationally tractable alternative.

## 4   Simulations

We compare two parametric models (independent Bernoulli and cascaded logistic model) with three nonparametric models (two universal binary models centered on the parametric models, and a naive histogram estimator) on simulated data with 15 neurons. We find the MAP solution as the parameter estimate for each model. We use an $l_1$ regularization to fit the cascaded logistic model and the corresponding UBM. The $l_1$ regularizer $\lambda$ was selected by scanning on a grid until the cross-validation likelihood started decreasing on 10% of the training data.

In Fig. 3, we simulate a maximum entropy (maxent) distribution with a third order interaction. As the number of samples increases, Jensen-Shannon (JS) divergence between the estimated model and true maxent model decreases exponentially for the nonparametric models. The JS-divergence of the

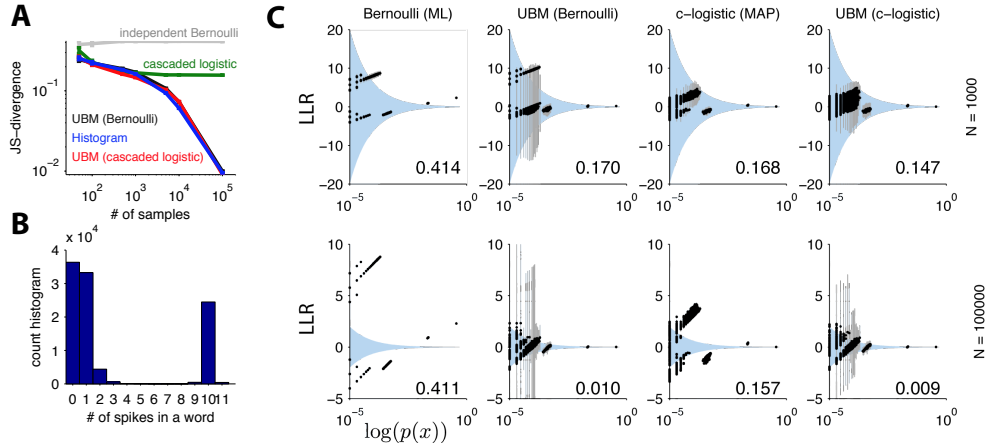

Figure 4: Synchrony histogram model. Each word with the same number of total spikes regardless of neuron identity has the same probability. Both Bernoulli and cascaded logistic models do not provide a good approximation in this case and saturate, in terms of JS divergence. Same format as Fig. 3.

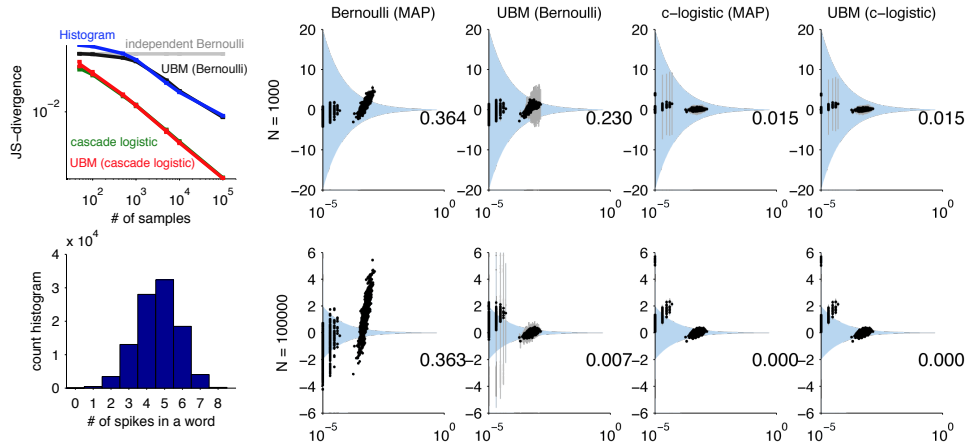

Figure 5: Ising model with 1-D nearest neighbor interaction. Same format as Fig. 3. Note that cascaded logistic and UBM with cascaded logistic base measure perform almost identically, and their convergence does not saturate (as expected by Theorem 1).

parametric models saturates since the actual distribution does not lie within the same parametric family. The cascaded logistic model and the UBM centered on it show the best performance for the small sample regime, but eventually other nonparametric models catch up with the cascaded logistic model.

The scatter plot (Fig. 3C) displays the log-likelihood ratio $\log(P_{\text{true}}) - \log(P_{\text{model}})$ to quantify the accuracy of the predictive distribution. Where significant deviations from the base measure model can be observed in Fig. 3C, the corresponding UBM adapts to account for those deviations.

In Fig. 4, we draw samples from a distribution with higher-order dependences; Each word with the same number of total spikes are assigned the same probability. For example, words with exactly 10 neurons spiking (and 5 not spiking, out of 15 neurons) occur with high probability as can be seen from the histogram of the total spikes (Fig. 4B). Neither the Bernoulli model nor the cascaded logistic model can capture this structure accurately, indicated by a plateau in the convergence plots (Fig. 4A,C). In this case, all three nonparameteric models behave similarly: both UBMs converge with the histogram.

In addition, we see that if the data comes from the model class assumed by the base measure, then UBM is just as good as the base measure alone (Fig. 5). Together, these results suggest that UBM

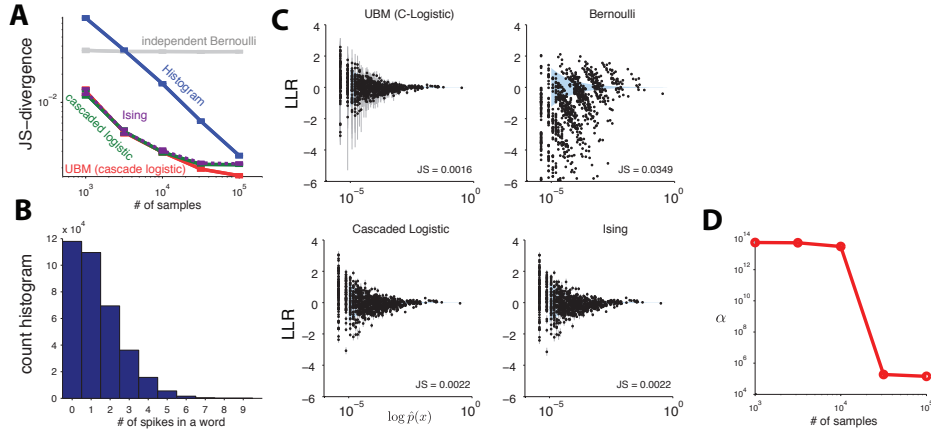

Figure 6: Various models fit to a population of ten retinal ganglion neurons' response to naturalistic movie [3]. Words consisted of 20 ms, binarized responses. $1 \times 10^5$ samples were reserved for testing. **(A)** JS divergence between the estimated model, and histogram constructed from the test data. Ising model is included, and its trace is closely followed by the cascaded logistic model. **(B)** Histogram of number of spikes per word. **(C)** Log-likelihood ratio scatter plot for the models trained with $10^5$ randomized observations. **(D)** The concentration parameter $\alpha$ as a function of sample size.

supplements the base measure to model flexibly the observed firing patterns, and performs at least as well as the histogram in the worst case.

## 5   Neural data

We apply UBMs to a simultaneously recorded population of 10 retinal ganglion cells, and compare to the Ising model. In Fig. 6A we evaluate the convergence of each model. Three models—cascaded logistic, its corresponding UBM, and the Ising model—initially perform similarly, however, as more data is provided, UBM predicts the probabilities better. In panel C, we confirm that the cascaded logistic UBM gives the best fit. The decrease in corresponding $\alpha$, shown in panel D, indicates that the cascaded logistic UBM is becoming less confident that the data is from an actual cascaded logistic model as we obtain more data.

## 6   Conclusion

We proposed universal binary models (UBMs), a nonparametric framework that extends parametric models of neural recordings. UBMs flexibly trade off between smoothing from the base measure and "histogram-like" behavior. The Dirichlet process can incorporate deviations from the base measure when supported by the data, even as the base measure buttresses the nonparametric approach with desirable properties of parametric models, such as fast convergence and interpretability. Unlike the reliable interaction model [3], which aims to provide the same features in a heuristic manner, the UBM is a well-defined probabilistic model.

Since the main source of smoothing is the base measure, UBM's ability to extrapolate is limited to repeatedly observed words. However, UBM is capable of adjusting the probabilities of the most frequent words to focus on fitting the regularities of small probability events.

We proposed the cascaded logistic model for use as a powerful, but still computationally tractable, base measure. We showed, both theoretically and empirically, that the cascaded logistic model is an effective, scalable alternative to the Ising model, which is usually limited to smaller populations. The UBM model class has the potential to reveal complex structure in large-scale recordings without the limitations of *a priori* parametric assumptions.

## Acknowledgments

We thank R. Segev and E. Ganmor for the retinal data. This work was supported by a Sloan Research Fellowship, McKnight Scholar's Award, and NSF CAREER Award IIS-1150186 (JP).

## Footnotes

[1] Technically, the mode of $\boldsymbol{\pi}$ is $G_\theta$ only for $\alpha \geq 1$, and for $\alpha < 1$, the distribution is symmetric around $G_\theta$, but the probability mass is concentrated on the corners of the simplex.

[2] Also known as the *logistic autoregressive network*. See [15], chapter 3.2.

[3] Note that for $x_i \in \{0, 1\}$, the mean $h_i$'s can be incorporated as the diagonal of $J$.

[4]We provide MATLAB code to convert back and forth between a subset of Ising models and the corresponding subset of cascaded logistic models (see online supplemental material).

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
