[Supplementary Material · UBM_NIPS2013_revis1_supplement.pdf]

# A   Proof of theorem 1

**Theorem 1.** *An Ising model with $J_{ij} = 0$ for $j < i - 2$ or $j > i + 2$, is also a cascaded logistic model. Moreover, the parameter transformation is bijective.*

The matrix form of the cascaded logistic model is

$$
W = \begin{bmatrix}
h_1 & 0 & \cdots & & & & 0 \\
h_2 & w_{2,1} & 0 & \cdots & & & 0 \\
h_3 & w_{3,1} & w_{3,2} & 0 & \cdots & & 0 \\
h_4 & 0 & w_{4,2} & w_{4,3} & 0 & \cdots & 0 \\
& & & \ddots & & & \\
h_m & 0 & \cdots & & 0 & w_{m,m-2} & w_{m,m-1}
\end{bmatrix}
\tag{21}
$$

and the probabilities for each word can be written as

$$
P(x_{1:m}) = P(x_1)P(x_2|x_2)\prod_{i=1}^{m} P(x_i|x_{1:i-1})
\tag{22}
$$

$$
P(x_i|x_{1:i-1}) = P(x_i|x_{i-2}, x_{i-1}) = \frac{\exp(x_i(h_i + x_{i-1}w_{i,i-1} + x_{i-2}w_{i,i-2}))}{1 + \exp(h_i + x_{i-1}w_{i,i-1} + x_{i-2}w_{i,i-2})}.
\tag{23}
$$

The pentadiagonal Ising model parameters are

$$
J = \begin{bmatrix}
J_{1,1} & J_{1,2} & J_{1,3} & 0 & \cdots & & & & 0 \\
J_{2,1} & J_{2,2} & J_{2,3} & J_{2,4} & 0 & \cdots & & & 0 \\
J_{3,1} & J_{3,2} & J_{3,3} & J_{3,4} & J_{3,5} & 0 & \cdots & & 0 \\
0 & J_{4,2} & J_{4,3} & J_{4,4} & J_{4,5} & J_{4,6} & 0 & \cdots & 0 \\
& & & \ddots & & & & & \\
0 & & & & \cdots & 0 & J_{m-2,m} & J_{m-1,m} & J_{m,m}
\end{bmatrix}
\tag{24}
$$

The map between model parameters is

$$
J_{m,m} = h_m
\tag{25}
$$

$$
J_{m-1,m} = w_{m,m-1}
\tag{26}
$$

$$
J_{m-1,m-1} = h_{m-1} + \log\left(\frac{1 + \exp(h_m)}{1 + \exp(h_m + w_{m,m-1})}\right)
\tag{27}
$$

$$
J_{i,i} = h_i + \log\left(\frac{1 + \exp(h_{i+1})}{1 + \exp(h_{i+1} + w_{i+1,i})}\right) + \log\left(\frac{1 + \exp(h_{i+2})}{1 + \exp(h_{i+2} + w_{i+2,i})}\right)
\tag{28}
$$

$$
J_{i,i+1} = w_{i+1,i} + \log\left(\frac{(1 + \exp(h_{i+2} + w_{i+2,i}))(1 + \exp(h_{i+2} + w_{i+2,i+1}))}{(1 + \exp(h_{i+2}))(1 + \exp(h_{i+2} + w_{i+2,i+1} + w_{i+2,i}))}\right)
\tag{29}
$$

$$
J_{i,i+2} = w_{i+2,i}
\tag{30}
$$

for $i \in \{1, \ldots, m - 2\}$.

*Proof.* We show that the parameter mapping from a cascaded logistic model defines a maximum entropy model with second order interactions for the $m = 2$ case and use induction.

For the initial case, we check each probability.

$$P(x_1 = 0, x_2 = 0) = P(x_1 = 0)P(x_2 = 0|x_1 = 0) = \frac{1}{1 + \exp(h_1)} \frac{1}{1 + \exp(h_2)}$$

$$= \frac{1}{1 + \exp(h_1)} \frac{1}{1 + \exp(h_2)} \exp(0)$$

$$P(x_1 = 0, x_2 = 1) = P(x_1 = 0)P(x_2 = 1|x_1 = 0) = \frac{1}{1 + \exp(h_1)} \frac{h_2}{1 + \exp(h_2)}$$

$$= \frac{1}{1 + \exp(h_1)} \frac{1}{1 + \exp(h_2)} \exp(d_2)$$

$$P(x_1 = 1, x_2 = 0) = P(x_1 = 1)P(x_2 = 0|x_1 = 1) = \frac{\exp(h_1)}{1 + \exp(h_1)} \frac{1}{1 + \exp(h_2 + w_{2,1})}$$

$$= \frac{1}{1 + \exp(h_1)} \frac{1}{1 + \exp(h_2 + w_{2,1})} \frac{1 + \exp(h_2)}{1 + \exp(h_2)} \exp(h_1)$$

$$= \frac{1}{1 + \exp(h_1)} \frac{1}{1 + \exp(h_2)} \exp\left\{ h_1 + \log\left( \frac{1 + \exp(h_2)}{1 + \exp(h_2 + w_{2,1})} \right) \right\}$$

$$= \frac{1}{1 + \exp(h_1)} \frac{1}{1 + \exp(h_2)} \exp(d_1)$$

$$P(x_1 = 1, x_2 = 1) = P(x_1 = 1)P(x_2 = 1|x_1 = 1) = \frac{\exp(h_1)}{1 + \exp(h_1)} \frac{\exp(h_2 + w_{2,1})}{1 + \exp(h_2 + w_{2,1})}$$

$$= \frac{1}{1 + \exp(h_1)} \frac{1}{1 + \exp(h_2)} \frac{(1 + \exp(h_2)) \exp(h_1) \exp(h_2 + w_{2,1})}{1 + \exp(h_2 + w_{2,1})}$$

$$= \frac{1}{1 + \exp(h_1)} \frac{1}{1 + \exp(h_2)} \exp\left\{ h_1 + \log\left( \frac{1 + \exp(h_2)}{1 + \exp(h_2 + w_{2,1})} \right) + h_2 + w_{2,1} \right\}$$

$$= \frac{1}{1 + \exp(h_1)} \frac{1}{1 + \exp(h_2)} \exp(d_1 + d_2 + J_{1,2})$$

With $C = \frac{1}{1+\exp(h_1)} \frac{1}{1+\exp(h_2)}$, we conclude that the cascaded logistic is equivalent to the Ising model for the $m = 2$ case.

For the induction step, we assume that $P_*(x_{2:m})$ is cascaded logistic and the parameter mapping gives,

$$P_*(x_{2:m}) = C_{2:m} \exp\left\{ \sum_{i=2}^{m} J_{i,i} x_i + \sum_{i=2}^{m-1} J_{i,i+1} x_i x_{i+1} + \sum_{i=2}^{m-2} J_{i,i+2} x_i x_{i+1} \right\} \tag{31}$$

$$C_{2:m} = \prod_{i=1}^{m} \frac{1}{1 + \exp(h_i)} \tag{32}$$

We extend the sparse cascaded logistic model to $x_{1:m}$ (note the direction of the induction) so that

$$P(x_{1:m}) = P(x_1)P(x_2|x_1)P(x_3|x_2, x_1) \prod_{i=4}^{m} P(x_i|x_{i-1}, x_{i-2}) \tag{33}$$

When $x_1 = 0$, then

$$P(x_{1:m}|x_1 = 0) = P(x_1 = 0)P_*(x_{2:m})$$

$$= \frac{1}{1 + \exp(h_1)} P_*(x_{2:m})$$

$$= \frac{1}{1 + \exp(h_1)} C_{2:m} \exp\left\{ \sum_{i=2}^{m} J_{i,i} x_i + \sum_{i=2}^{m-1} J_{i,i+1} x_i x_{i+1} + \sum_{i=2}^{m-2} J_{i,i+2} x_i x_{i+1} \right\}$$

$$= C_{1:m} \exp\left\{ \sum_{i=1}^{m} J_{i,i} x_i + \sum_{i=1}^{m-1} J_{i,i+1} x_i x_{i+1} + \sum_{i=1}^{m-2} J_{i,i+2} x_i x_{i+1} \right\}$$

For the $x_1 = 1$ case

$$P(x_{1:m}|x_1 = 1) = P(x_1 = 1)P(x_2|x_1 = 1)P(x_3|x_2, x_1 = 1)\prod_{i=4}^{m}P(x_i|x_{i-1}, x_{i-2})$$

$$= \frac{\exp(h_1)}{1 + \exp(h_1)}\frac{\exp(x_2(h_2 + w_{2,1}))}{1 + \exp(h_2 + w_{2,1})}\frac{\exp(x_3(h_2 + x_2 w_{3,2} + w_{3,1}))}{1 + \exp(h_3 + x_2 w_{3,2} + w_{3,1})}\prod_{i=4}^{m}P(x_i|x_{i-1}, x_{i-2})$$

$$= \frac{\exp(h_1)}{1 + \exp(h_1)}\frac{\exp(x_2(h_2 + w_{2,1}))}{1 + \exp(h_2 + w_{2,1})}\frac{\exp(x_3(h_2 + x_2 w_{3,2} + w_{3,1}))}{1 + \exp(h_3 + x_2 w_{3,2} + w_{3,1})}$$

$$\cdot \frac{P_*(x_2)P_*(x_3|x_2)}{P_*(x_2)P_*(x_3|x_2)}\prod_{i=4}^{m}P(x_i|x_{i-1}, x_{i-2})$$

$$= \frac{\exp(h_1)}{1 + \exp(h_1)}\frac{\exp(x_2(h_2 + w_{2,1}))}{1 + \exp(h_2 + w_{2,1})}\frac{\exp(x_3(h_3 + x_2 w_{3,2} + w_{3,1}))}{1 + \exp(h_3 + x_2 w_{3,2} + w_{3,1})}$$

$$\cdot \frac{1 + \exp(h_2)}{\exp(x_2(h_2))}\frac{1 + \exp(h_3 + x_2 w_{3,2})}{\exp(x_3(h_3 + x_2 w_{3,2}))}P_*(x_{2:m})$$

$$= \frac{\exp(h_1)}{1 + \exp(h_1)}\exp\left\{\log\left(\frac{1 + \exp(h_2)}{1 + \exp(h_2 + w_{2,1})}\right) + x_2 h_2 + x_2 w_{2,1}\right.$$

$$+ \log\left(\frac{1}{1 + \exp(h_3 + w_{3,1})}\right) + x_2 \log\left(\frac{1 + \exp(h_3 + w_{3,1})}{1 + \exp(h_3 + w_{3,2} + w_{3,1})}\right)$$

$$+ x_3 h_3 + x_2 x_3 w_{3,2} + x_3 w_{3,1} - x_2 h_2$$

$$\left. + \log\left(1 + \exp(h_3)\right) + x_2 \log\left(\frac{1 + \exp(h_3 + w_{3,2})}{1 + \exp(h_3)}\right) - x_3 h_3 - x_2 x_3 w_{3,2}\right\}$$

$$\cdot P_*(x_{2:m})$$

$$= \exp\left\{x_1\left(h_1 + \log\left(\frac{1 + \exp(h_2)}{1 + \exp(h_2 + w_{2,1})}\right) + \log\left(\frac{1 + \exp(h_3)}{1 + \exp(h_3 + w_{3,1})}\right)\right)\right.$$

$$\left. + x_1 x_2\left(w_{2,1} + \log\left(\frac{1 + \exp(h_3 + w_{3,1})}{1 + \exp(h_3 + w_{3,2} + w_{3,1})}\right)\right) + x_1 x_3 w_{3,1}\right\}$$

$$\cdot \frac{1}{1 + \exp(h_1)}P_*(x_{2:m})$$

$$= \exp\left\{x_1 d_1 + x_1 x_2 J_{1,2} + x_1 x_3 J_{1,3}\right\}$$

$$\cdot \frac{1}{1 + \exp(h_1)}C_{2:m}\exp\left\{\sum_{i=2}^{m}J_{i,i}x_i + \sum_{i=2}^{m-1}J_{i,i+1}x_i x_{i+1} + \sum_{i=2}^{m-2}J_{i,i+2}x_i x_{i+1}\right\}$$

$$= C_{1:m}\exp\left\{\sum_{i=1}^{m}J_{i,i}x_i + \sum_{i=1}^{m-1}J_{i,i+1}x_i x_{i+1} + \sum_{i=1}^{m-2}J_{i,i+2}x_i x_{i+1}\right\}$$

The reverse direction, mapping the banded Ising model into a banded cascaded logistic model, follows from the fact that the parameter mapping is invertible. $\square$

# B  Second derivatives of UBM

The second derivatives of the log-likelihood (6) are

$$\frac{\partial^2 L}{\partial\theta_i \partial\theta_j} = \alpha^2 \sum_k \left(\psi_1(n_k + \alpha g_k) - \psi_1(\alpha g_k)\right)\frac{\partial g_k}{\partial\theta_i}\frac{\partial g_k}{\partial\theta_j} + \alpha \sum_k \left(\psi(n_k + \alpha g_k) - \psi(\alpha g_k)\right)\frac{\partial^2 g_k}{\partial\theta_i \partial\theta_j} \quad (34)$$

$$\frac{\partial^2 L}{\partial^2 \alpha} = \sum_k g_k^2 \left(\psi_1(n_k + \alpha g_k) - \psi_1(\alpha g_k)\right) + \psi_1(\alpha) - \psi_1\left(N + \alpha\right), \quad (35)$$

where $\psi_1(\cdot)$ is the trigamma function. Since $\psi_1$ is a monotonically decreasing function, the first term is negative, while the second term is positive.