[Reviews · NeurIPS 2013]

Submitted by Assigned_Reviewer_2

This paper presents an elegant nonparametric generalization of widely-used parametric neural population models. This generalization is able to capture statistical patterns that diverge from the parametric model predictions. Interestingly, they also show that two apparently different models (Ising and cascaded logistic) are equivalent to each other under certain conditions.

QUALITY: The models and analyses are rigorous and the experiments seem to be fairly thorough.
- My main substantive comment is that it seems weird to model the firing patterns of a neural population without taking into account the inputs. For example, the data analyzed in section 5 reflect responses to gabors that vary in orientation, but the dependence on orientation is not modeled. I realize that this may be conventional in this literature, but it still seems wrong to me.
- Should cite something for the Dirichlet process.
- p. 4: Is Eq. 11 correct? p(x_i=0|x_{1:i-1}) doesn't equal 0 but rather 1-p(x_i=1|x_{1:i-1}).

CLARITY: The paper is clearly written.

ORIGINALITY: The model and results in this paper are original.

SIGNIFICANCE: This paper will be of significance to neuroscientists working on modeling neural populations.

MINOR COMMENTS:
- p. 3: "for which we" -> "when we"
- p. 4: the Ising model's partition function does have a closed form (since it's discrete), it's just usually intractable.
Summary: A well-written paper with some interesting theoretical results. Of interest mainly to neuroscientists working on neural population modeling.

Submitted by Assigned_Reviewer_5

Review for #1163 “Universal models for binary spike patterns using centered Dirichlet processes.”

The goal of this paper is to provide a more accurate method for modeling the distribution of binary spike patterns over a population of neurons. Essentially what the authors are trying to do is improve upon parametric models of the pattern distribution (such as a Bernoulli, cascaded logistic or Ising model) by allowing for deviations from the parametric model (or base model) if they are justified by the data. The involves postulating a Dirichlet process centered upon the base model and fitting the parameters of the base model and the concentration parameter of the Dirichlet process via gradient ascent (although I imagine other methods could be used for fitting). Intuitively this constitutes fitting a type of weighted average between the probability distribution of the base model and the pattern probabilities estimated by counting alone. Thus the base model provides a smoothing (over the naïve, counting based maximum likelihood estimate) and the weighting with the counting based estimate allows for deviations from this distribution if they are strong enough in the data. The authors discuss several tractable base models (Bernoulli and cascaded logistic) and then proceed to demonstrate that their method optimally captures (when compared to the base models alone and also counting based estimates) the pattern probability distribution for both simulated data and for a population of 10 V1 neurons.

I enjoyed reading this paper. It was very clear what the Authors were attempting to do and estimating pattern probability distributions that deviate from standard parametric models (Ising, GLM etc.) is an important goal. The formalism itself seems to be rather standard (very similar to Chapter 25 of Murphy’s Machine Learning book) but I think the application is new, or at least I’ve never seen anyone apply such methods to neural data before. In this vein, I would have liked to have seen more application to real data … but I do understand the space constraints of a NIPS paper. If the authors can, in the final version, include another real data example I think that would make their paper more appealing … but I won’t insist upon this, its just a recommendation.

I don’t have any major corrections to the paper. I would recommend some minor additions to the introduction or discussion (a sentence or two would suffice) motivating what the Author’s method is useful for. If I understand the paper correctly, it seems to me that the main utility of this work is in identifying the existence of deviations from the base model as opposed to explaining the source of those deviations. Parametric base models have explanatory power, for example a well-fit Ising model indicates that second order correlations are sufficient to describe the population’s spiking statistics. The Dirichlet process, being non-parametric, really doesn’t tell you why the distribution deviates from the base model … just that it does. If I’m correct here, I think the authors should put a sentence or two in the discussion about these points.

There are also a couple typos so a proofreading would be good. The most notable is that the last sentence of the introduction ends mid sentence.

In summary, while the mathematical formalism isn’t new, the application is novel and interesting. I think the paper will be of use to the computational neuroscience community and should be accepted as a NIPS paper subject to some minor edits and proofreading.


Summary: A nice clear presentation of a method for determining if the probability distribution of spike patterns (across a neural population) differs from a parametric "base" model. The method uses a standard Dirichlet process centered upon neurophysiological base models to generalize these models if the deviations from the base are strong enough in the data. While the mathematics are not exactly new, the application is new to my knowledge and quite interesting. This paper should be accepted with minor edits.

Submitted by Assigned_Reviewer_6

This paper combines two main ideas: 1- an interesting proposal for a new nonparametric spike train model, and 2- an equivalence between certain logistic regression models and Ising (maximum entropy) spike train models.  The two pieces don't entirely hang together, and both parts could be fleshed out some more; the paper as a whole feels like two not-quite-complete papers that have been shoved together to make a longer paper.

There are also a number of typos that should be corrected.

That aside, some more substantive comments. Re: the first part - I agree about the importance of introducing new, more flexible spike train models.  It's clear that there's more to life than logistic regression models and Ising models.  This paper takes a good step in the right direction, and demonstrates that incorporating a nonparametric component is useful. (It would be worth citing Sam Behseta here, who has done some relevant work in this area recently.) That said, the authors don't manage to go beyond this and actually show that the new model can do something new for us, or lead to some novel insight about population coding. But maybe that's too much to expect in a nips paper. It should also be noted that there are many alternative ways of moving beyond the simple logistic-type models. Dirichlet mixtures of logistics would be another possible route. Some more discussion about alternative possibilities would be useful.

Minor - is the predictive distribution (5) ever used? if not, it doesn't seem to add much here.

More major - the authors propose a kind of empirical bayes approach for estimating the model - they optimize the marginal likelihood of the data as a function of the base measure parameters \alpha and \theta. I can't quite tell if the posterior estimated in this way will be consistent, in the sense that
p(X| \hat \alpha, G_{\hat \theta})
converges to a delta function at the true data-generating distribution, where \hat \alpha and \hat \theta denote the parameters estimated via the maximum marginal likelihood procedure described in sec 2.1. It would be great if the authors could clarify this. (Clearly the statement is true for fixed \alpha and \theta - but what happens when \alpha and \theta adapt to the data as well?)

A note about scalability - the authors claim the proposed methods are highly scalable, but I worry a bit about the non-convexity of their marginal likelihood. Certainly if we use a complicated, high-dimensional model for \theta we can run into trouble here. This argues against the scalability of the method. Worth discussing a bit.

The cascaded logistic model is computationally easy because neuron i's rate only depends on the firing of neurons 1:i-1. So the order in which these neurons are arranged is important - neuron i has i parameters to adjust, which means that neuron 1 seems much less flexible than neuron m. Maybe there's some reason why the ordering turns out not to matter here? If not, then how do we choose the "right" order? This issue needs to be addressed.

Fig 5 caption - i didn't understand what was meant by "perform … perfectly" here.

It's unclear what conclusion to draw from sec 5 / fig 6, beyond "our code can be run on real data, too."
Summary: This paper combines two main ideas: 1- an interesting proposal for a new nonparametric spike train model, and 2- an equivalence between certain logistic regression models and Ising (maximum entropy) spike train models. The two pieces don't entirely hang together, and both parts could be fleshed out some more; the paper as a whole feels like two not-quite-complete papers that have been shoved together to make a longer paper.
Author Feedback

Author rebuttal: We thank the reviewers for their detailed and thoughtful comments. We
greatly appreciate the time and effort of all reviewers, and believe
their suggestions will substantially improve our manuscript. We will
first briefly address points raised by Reviewers #2 and #5, and then
address the comments of Reviewer #6 in greater detail.

Reviewer #2:
----------

Thank you; we will add references on DP and correct equation
#11. Also, we agree about the importance of stimulus-conditional
distributions; our work follows others (e.g., using the Ising model)
that focus primarily on marginal distributions, though we should point
out that we can also use the model to describe spike patterns
conditioned on fixed (discrete) stimuli, making it suitable (e.g.) for
decoding analyses.

Reviewer #3:
----------

Thank you for the enthusiastic review. We will attempt to add another
neural data example and revise the Discussion as suggested.

Reviewer #6:
----------

1. Nonparametric and Cascaded-Logistic vs Ising pieces not hanging
together:

We agree that the comparison between Ising and the cascaded logistic
model could make an independent story. However, since the UBM isn't
scalable without a scalable base measure, we view the cascaded
logistic model as an essential part of the current paper (see also
comment 6). We will make this connection more clear in our writing.

2. Citation to Behseta's work: thanks, we will add this and citations
to other relevant literature on NP Bayes methods in neuroscience.

3. Mixture models

Mixture modeling is definitely a viable alternative for modeling
distributions. We are aware that Bernoulli mixture models have been
used in modeling binary images, for example. We thank the reviewer for
the suggestion, and plan to modify the paper to include discussion
along these lines.

4. When do we use predictive distribution (5)?

The scatter plots and Jensen-Shannon divergences in the results
(Figs. 3 to 6) are computed with the predictive distribution. We will
clarify this in the paper.

5. Does the posterior concentrate with the empirical Bayes like MAP
inference procedure on (alpha, theta)?

Thank you for raising this important issue. Since the posterior
concentrates on the true distribution for fixed \theta and \alpha as
the reviewer noted, it should not (in general) perform worse when
\theta and \alpha are adapted to the data. (That is, a random setting
of hyperparameters shouldn't do better than a maximum likelihood
setting of the hyperparameters.) We concede that we haven't proved
posterior concentration however. For now, we can guarantee only that
the posterior concentrates when \alpha remains finite (since \alpha
determines the total contribution of the base measure, which will be
overwhelmed by the data so long as alpha is finite.)

In practice, what we observe is that if data truly come from the base
distribution, alpha runs off to infinity as sample size increases,
leaving a parametric model (which is the correct model in this case).
By contrast, if data come from a distribution not covered by the
parametric base measure, alpha converges to zero with more data,
resulting in a pure "histogram" model in the limit of infinite
data. In both scenarios, the posterior concentrates on the true
distribution.

6. Scalability: if we use a complicated, high-dimensional model for
theta we can run into trouble here.

We have shown only that the UBM is scalable for the Bernoulli and
cascade-logistic base measures (the latter of which is certainly
high-dimensional, though we don't regard it as complicated due to its
convexity and the ease of fitting and normalizing it). However, the
UBM approach scales for any normalizable base measure with convex
negative log-likelihood. Non-convexity of the joint likelihood appears
only in the alpha parameter, as shown in the supplement (Eqs. 34 and
35). This means that we can tractably perform a 1D line search for
\alpha (where \theta is generally high-dimensional).

7. Importance of ordering in cascaded logistic model:

We agree this is an important point. On page 5 we suggested using the
Cuthill-McKee algorithm as a possible solution to the ordering; we
actually spent some time experimenting with orderings based on this
algorithm, but found we obtained almost identically good fits even
with random orderings, and so elected to leave this out of the
paper. But we do plan to explore the issue of how/whether ordering
matters in future work.

8. Fig 5 caption - I didn't understand what was meant by "perform ...
perfectly" here.

We apologize for our lack of clarity. We know theoretically that
cascaded logistic is in the correct model class for Fig 5., and what
we intended to say was that the convergence does not saturate as in
other examples.

We will address these issues in the final manuscript. Thanks again
for the detailed and constructive comments.